# Promoting Physical Activity among Working Women: The Influence of Perceived Policy Effectiveness and Health Awareness

**DOI:** 10.3390/ijerph20021021

**Published:** 2023-01-05

**Authors:** Huilin Wang, Ziqing Xu, Jingyu Yang, Dan Huang

**Affiliations:** 1School of Business, Hunan University of Science and Technology, Xiangtan 411201, China; 2Faculty of Economics, Chulalongkorn University, Bangkok 10330, Thailand; 3International College, National Institute of Development Administration, Bangkok 10240, Thailand; 4Department of Medical Bioinformatics, University of Göttingen, 37077 Göttingen, Germany; 5School of Physical Education, Hunan University of Science and Technology, Xiangtan 411201, China

**Keywords:** working women, physical activity, perceived policy effectiveness, health awareness

## Abstract

In recent years, patients with chronic diseases have shown a younger trend due to physical inactivity and irregular lifestyles. Accordingly, the Chinese government has implemented the “National Fitness Program”, which aims to enhance people’s health by popularizing exercise and a healthy lifestyle. However, women are less physically active than men, and how to appeal to women to devote themselves to fitness activities has become a social concern. Based on the expanded theory of planned behavior (TPB), this study explores the impact of the perceived policy effectiveness and health awareness on physical activity among working women. This study adopted a repeated cross-sectional study method, and each respondent was asked to complete a two-stage survey. The structural model of the extended TPB was tested using sample data from 376 working women in Changsha, China. The results show that perceived policy effectiveness and health awareness positively affect actual behavioral control and implementation intention. Among them, perceived policy effectiveness has the most significant impact on implementation intention, followed by health awareness. Furthermore, actual behavioral control and implementation intention mediate the relationship between perceived policy effectiveness/health awareness and physical activity. The findings suggest that to promote physical activity among working women, the Chinese government should deepen the implementation of the “National Fitness Program” and raise the public’s health awareness.

## 1. Introduction

Because of high pressure, frequent overtime and staying late, lack of necessary exercise, sedentariness, and irregular diet, women in urban workplaces experience different degrees of sub-health [1]. It is worth noting that the sub-health state has become common for women born in the 1960s and 1990s in China. Sleep problems, gastrointestinal discomforts, skin problems, cervical and lumbar pains, and menstrual irregularities are the health problems most frequently mentioned by working women [2]. Over time, more and more patients with chronic diseases, such as cancer, heart disease, stroke, and diabetes, tend to be younger [3,4]. As a result, chronic diseases have accounted for 88.5% of deaths [5]. Furthermore, physical inactivity is the leading risk factor for mortality globally, further contributing to the chronic disease epidemic [6]. Not only does this increase the pressure on national public health services, but it may also lead to a potential decline in life expectancy for future generations [3].

One in three adults worldwide is physically inactive [7]. Especially in East and Southeast Asia, women are less physically active than men [8]. Several studies have shown that physical activity can reduce obesity [9], chronic disease [10], and mental health problems [11], as well as promote health [12], reduce stress [13], and improve self-efficacy [14]. Accordingly, the Chinese government has introduced a series of policies to improve its citizens’ health; the most famous is the “National Fitness Program” [15]. This program not only calls on the public to participate in physical activities through policy propaganda but also provides material conditions and guarantees for the public to participate in physical activities [13]. However, the impact of the “National Fitness Program” on physical activity among working women has not been discussed by academics. Therefore, it is of theoretical and practical significance to study how to encourage working women to avoid inactivity and pursue a healthy lifestyle by implementing the “National Fitness Program”.

Most studies focusing on physical activity tend to explore the factors influencing people’s physical activity [16] and the benefits of physical activity [17]. However, among the factors affecting physical activity, perceived policy effectiveness and health awareness are the most easily overlooked by researchers. Based on an expanded TPB, this study is the first to discuss the impact of perceived policy effectiveness and health awareness on people’s physical activity. In addition, the researchers upgraded the content of the original model of the TPB, such as using actual behavioral control that reflects the individual’s behavioral control ability more realistically than perceived behavioral control, and implementation intention, which has a more specific plan than behavioral intention. This research design enriches the theoretical body of research on the TPB and physical activity. Therefore, the objectives of this study are as follows: (1) to understand the physical activity of working women under the influence of the “National Fitness Program”; (2) to analyze the factors that affect the physical activity of working women; and (3) to provide advice to relevant agencies (e.g., policymakers, local communities, businesses).

A repeated cross-sectional study method was adopted in this study through two stages of data collection to best truly reveal the behavioral mechanisms of individuals. The results show that perceived policy effectiveness and health awareness significantly positively affect actual behavioral control and implementation intention. In addition, actual behavioral control and implementation intention mediate the relationship between perceived policy effectiveness/health awareness and physical activity. The findings indicate that working women with high perceptions of policy effectiveness, strong health awareness, strong actual behavioral control, and firm implementation intention are more active in exercising. This study provides rich information for developing China’s “National Fitness Program” through an in-depth analysis of the external environmental and psychological factors that affect the physical activity of working women. The findings of this study also advance multidisciplinary applications in public policy, sports science, and behavioral psychology.

## 2. Literature Review

### 2.1. Theory of Planned Behavior (TPB)

Physical activity is a behavior driven by a complex decision-making process [18], and there have been attempts in the academic area to promote the study of physical activity. However, previous research may not be sufficient to explain people’s physical activity, ignoring people’s psychological motivation and situational factors, which may be challenging to reveal the fundamental driving factors behind people’s physical activity. The TPB is a well-known attitude-behavior relationship theory in social psychology [19]. It believes that people engage in behavior under conscious choices and plans, focusing on attitudes, subject norms, and perceived behavioral control for the intention and behavior.

According to the TPB, physical activity results from the individual’s active choice and is influenced by several factors. In this process, the implementation intention to participate in physical activity is the key [20]. Therefore, analyzing the antecedent variables that affect the implementation intention and the factors that affect the transformation of behavioral intention to behavior is particularly important for understanding the process of physical activity. Although existing studies have paid increasing attention to implementation intention and physical activity, current research lacks a systematic and comprehensive theoretical framework to reveal the antecedent variables of implantation intention. In promoting China’s “National Fitness Program”, the State Council of China has issued a series of policies to support and encourage working women to participate in more physical activity. At the same time, with the increasing pressure on working women’s work and family life, their health awareness is also increasing. Therefore, this study constructs an integrated model from perceived policy effectiveness and health awareness to physical activity and systematically reveals the theoretical framework of working women’s participation in physical activity.

### 2.2. National Fitness Program

The “National Fitness Program” of developed countries provides a reference for China. For example, the national physical activity in the UK has been around for 70 years. The British government has made it the highest goal of national fitness for citizens to develop the habit of sports participation. A management method combining “official” and “civilian” is adopted in implementing the fitness plan [21]. The UK has used government grants, sports lotteries, and charitable donations to raise funds. The British government also attaches great importance to promoting female physical activity. In 1992, the UK promulgated the “Women and Sports Policy”, which, for the first time, required women’s physical activity in policy, and then they successively launched many guidelines for women’s physical activity. For example, “Tennis Tuesdays” by The Lawn Tennis Association of Women in Sport and “This Girl Can” by the Leicester–Shire and Rutland Sport (LRS) [22]. It has effectively promoted women’s physical activity and provided many opportunities for women to participate in physical activity.

The promotion of national fitness in China began in the 1990s. Since the introduction of the first “National Fitness Program” in 1995, it has experienced more than 20 years of development. In 2021, the State Council launched the “National Fitness Program (2021–2025)”. The purpose is to solve the problems of the unbalanced regional development of national fitness and insufficient supply of public services. The policy pointed out that by 2025, the public service system for national fitness will be perfect, and people’s physical activity will be more convenient. As a result, the proportion of people who regularly participate in physical activity will reach 38.5%, and there will be 2.16 social sports instructors per 1000 people. In addition, it will build or renovate more than 2000 fitness venues and facilities, replenish more than 5000 townships’ (streets’) national fitness venues and equipment, and digitally upgrade and transform more than 1000 public sports venues [15]. These policies have also encouraged people to participate in physical activities.

## 3. Hypotheses

### 3.1. Perceived Policy Effectiveness, Actual Behavioral Control, and Implementation Intention

Physical activity is considered a meaningful way to cultivate public health self-management awareness and promote physical and mental health. Therefore, many countries have encouraged physical activity as an essential part of the public health agenda and have launched a variety of interventions to promote physical activity [23]. Among them, the implementation of physical activity policies has announced the promotion of public physical activities, the construction of public physical facilities, the promotion of sports events, and the dissemination of health knowledge. This study defines perceived policy effectiveness as an individual’s favorable or unfavorable evaluation of policy measures’ clarity, adequacy, and facilitation.

According to the TPB, perceived behavioral control is an individual’s perceived degree of control over their behaviors [24]. Unlike perceived behavioral control, actual behavioral control can better reflect the situation [25]. Actual behavioral control reflects the existing available resources and opportunities for individuals. This study is mainly reflected in the public fitness venues, fitness facilities, fitness activities, and the distance between the media and their homes, which will affect their behaviors to a certain extent. Numerous studies confirm that if a person has enough actual control over what they want to do, their intentions can be achieved. In the study of physical activity, a good sports environment positively affects sports participation. Perceived land use diversity [26], the distance to a public open space [12], the distance to an exercise destination [27], the environment of the fitness venues [28], satisfaction with the public facilities [29], and other factors will affect residents’ intention to do physical activities. In other words, policies encouraging physical activities can effectively promote residents’ actual behavioral control and intention to exercise. This study, thus, proposes the following hypotheses:

**Hypothesis** **1 (H1).**
*Perceived policy effectiveness has a positive impact on actual behavioral control.*


**Hypothesis** **2 (H2).**
*Perceived policy effectiveness has a positive impact on implementation intention.*


### 3.2. Health Awareness, Actual Behavioral Control, and Implementation Intention

Health awareness refers to the degree to which an individual’s emphasis and health concerns are integrated with daily life. It is a variable based on the degree to which individuals participate in healthy life choices. It is generally believed that individuals with health awareness are “health-oriented”. Many studies suggest that if individuals have health awareness, they will have a positive attitude toward choosing a healthy lifestyle.

According to the TPB, attitude positively affects people’s perceived behavioral control and executive intention. Lin et al. [30] found that working women are a group with solid anxiety about health and generally have a supportive attitude towards healthy living. According to Grimm et al. [31], increased work and family stress may threaten health. Individuals with high health awareness are more aware of and pay attention to their health status and changes. Accordingly, they may be more motivated to maintain or improve their health status and, thus, more willing to participate in physical activities.

On the other hand, individuals with higher health concerns are more sensitive to the adverse health effects of work stress [32]. According to the current working situation of working women in China, women are under increasing pressure from work, family, and society, which seriously impacts people’s lives and health [33]. Therefore, people’s motivation to participate in physical activities may be due to health considerations and concerns. In other words, when individuals are more aware of their health, their actual behavioral control and intention to perform physical activities will also be more robust. This study, thus, proposes the following hypotheses:

**Hypothesis** **3 (H3).**
*Health awareness has a positive impact on actual behavioral control.*


**Hypothesis** **4 (H4).**
*Health awareness has a positive impact on implementation intention.*


### 3.3. The Mediating Effects

Some studies have pointed out that the theoretical model of planned behavior explains the behavior mechanism to a certain extent. However, for physical activity, existing studies have found that factors such as behavioral habits [34], self-efficacy [35], and environment [36] play a moderating or mediating role in the process of converting health awareness into behavior, which is of great value in explaining and predicting behavior.

According to the TPB, exogenous factors indirectly influence intention through the personal situation perceptions of actual behavioral control [24]. Nguyen [37] argues that behavioral control determines implementation intention. The actual behavioral control is a product of the combined effects of several other exogenous variables, such as policy, awareness, and external elements. Compared with general intentions, implementation intentions have an obvious goal orientation for the actions to be performed and when, where, and how to carry out the intentions. For example, in the case of physical activity, the implementation intention can be stated as “I plan to run in the park next to my house from 6:00 AM to 7:00 AM before work every day”. Here, implementation intentions are associated with opportunities to act (to park next to home and time) with behavior (e.g., running, playing, dancing). Oteng–Peprah et al. [38] believe that people’s behavior will be driven by intention before it occurs. Whether an individual performs this behavior, the behavioral intention has an important impact on future behavior. For example, when individuals want to improve their physical condition, they will be more willing to inquire about relevant health advice to answer their doubts. In physical activity, actual behavioral control is the individual’s perception of how easy or difficult it is to perform a specific physical activity behavior. For example, after getting off work, you go to the next basketball court to play, which means that the resources needed by the individual to complete the goal of playing are relatively easy to obtain. Potthoff et al. [39] found that implementation intention interventions had a small to medium effect on objectively assessed health behavior. Therefore, this study thus proposes the following hypotheses:

**Hypothesis** **5 (H5).**
*Implementation intention mediates the positive effect on the relationship between actual behavioral control and physical activity.*


**Hypothesis** **6 (H6).**
*Actual behavioral control and implementation intention positively mediate the relationship between perceived policy effectiveness and physical activity.*


**Hypothesis** **7 (H7).**
*Actual behavioral control and implementation intention positively mediate the relationship between health awareness and physical activity.*


A summary of all the hypotheses is shown in Figure 1.

## 4. Methodology

### 4.1. Sampling

This study adopted the cluster sampling method. From June 2022 to July 2022, the researchers randomly selected 39 government agencies and enterprises in five municipal districts of Changsha City, China, and distributed questionnaires to women aged 18–55 working there. To ensure the randomness of the sampling process and the sample’s representativeness, the 39 government agencies and enterprises were evenly distributed in various areas of Changsha City, and the types of enterprises involved multiple types of large, medium, small, and micro. The questionnaire survey consisted of two stages. In the first stage, respondents were asked to complete an initial round of questionnaires on perceived policy effectiveness, health awareness, and implementation intention. One week later, the respondents were again invited to participate in the second stage of the survey, giving feedback on their actual behavioral control and physical activity during the week. In the first stage, 800 questionnaires were distributed, 662 valid questionnaires were retrieved, and the recovery rate of valid questionnaires was 82.8%. In the second stage, 662 questionnaires were distributed, and 376 valid questionnaires were recovered. As a result, the recovery rate of valid questionnaires was 56.8%. From this, the sample loss rate from the first to the second stage was calculated to be 43.2%.

### 4.2. Questionnaire

The questionnaire consists of six parts. Part 1 asked for information about the demographics of the respondents. The second part collected respondents’ perceived effectiveness of the “National Fitness Program”, a scale developed by Wang and Aweewan [25]. The third part collected data on health awareness among working women, a measure from Gould [40]. The fourth part collected data on working women’s implementation intentions to participate in physical activity, measured using a scale from Budden and Sagarin [41]. The fifth part collected data on the actual behavioral control of working women over the past week, measured using a scale developed by Wang and Aweewan [25]. Finally, the sixth part collected data on working women’s participation in physical activity in the past week, a scale developed by Andersen et al. [42]. All of the above items were measured using a five-point Likert scale (i.e., 1 = strongly disagree, 5 = strongly agree, or 1 = never, 5 = always).

Considering the specific research field and cultural background, the researchers revised parts of the scales, so pilot testing was also necessary to ensure the reliability of the revised scales [43]. Using convenience sampling, the researchers distributed 70 questionnaires to women living in a community and recovered 62 valid questionnaires. The pilot test results showed that the Cronbach coefficients were all higher than 0.8; thus, the measurement tools had good internal consistency.

### 4.3. Data Analysis

This study constructed a structural equation model (SEM) of physical activity among working women based on AMOS v.23. It used the maximum likelihood (ML) estimation method to estimate the model’s parameters. A two-step modeling approach was used to evaluate the measurement and structural models [44]. The model’s construct validity was first comprehensively assessed, and then the hypothetical model’s fitting coefficients and path coefficients were measured.

## 5. Results

### 5.1. Respondents’ Socio-Economic Characteristics

The background of the respondents is shown in Table 1. Regarding age composition, working women aged 29–44 account for about half of the total. Regarding education, 35.6% of working women had a high school/vocational school degree, followed by 32.7% with a college/university degree. Regarding occupation, 30.3% worked in the public sector, and 69.7% worked in the private sector. Finally, regarding monthly salary, 64.6% of working women earned less than 10,000 CNY (1500 USD) per month. Comparing the primary data reported in Changsha’s 2020 census, respondents’ socioeconomic background (e.g., age, education level, occupation, monthly salary) matched Changsha’s broader population, so the sample was somewhat representative.

### 5.2. Measurement Model

Reliability analysis included tests for Cronbach’s alpha coefficient and the composite reliability (CR) coefficient for the latent variables [47]. As shown in Table 2, Cronbach’s alpha coefficients for all the variables are in the range of 0.848 to 0.924, and the CR coefficients are in the range of 0.853 to 0.925, indicating that the test results can reflect the natural characteristics of the subjects. Convergent validity tests included factor loadings and the extracted mean variance (AVE) [47]. All the standardized factor loadings in Table 2 range from 0.742 to 0.912, and the AVEs in all the variables range from 0.571 to 0.805, all higher than the baseline value. The measurement of discriminant validity requires verifying the relationship between the correlation coefficient of each latent variable and the square root of the AVE. The square root value of the AVE for all the variables is more significant than the correlation coefficient between the variables (see Table 3), indicating that each variable has good discriminant validity.

### 5.3. Common Method Variance

Potential problems with the common method variance (CMV) are widespread in behavioral research, especially those using questionnaires [48]. First, this study conducted the CFA factor test to check for the presence of the CMV [49]. The results showed that the ratio of the substantially explained variance to the method-based mean variance (i.e., R_1_^2^/R_2_^2^) was approximately 1:1, proving the existence of the CMV. Next, the unmeasured latent method construct (ULMC) was used to estimate whether the presence of the CMV affected the model’s outcome [50]. The results showed that the p-value for the nested model comparison was 0.926, which was not significantly different, indicating that the CMV did not affect the model. Therefore, the model in this study did not need the CMV correction.

### 5.4. Structural Path Model

Since the standardized coefficients did not exceed 1, the error variance was positive, and no extremely large or small standard errors were generated, so the model did not generate violation estimates. In addition, the model fit index (χ^2^/df = 2.220, GFI = 0.918, NFI = 0.937, CFI = 0.964, TLI = 0.958, and RMSEA = 0.057) conformed to the standard, indicating that the overall fit of the model was good [51]. Table 3 shows a significant positive correlation among the independent, mediating, and dependent variables, which provided preliminary support for validating the hypotheses. The structural path model is shown in Figure 2; the effect of perceived policy effectiveness on actual behavioral control was statistically significant (*β* = 0.459, *p* < 0.001), supporting H1; the effect of perceived policy effectiveness on implementation intention was statistically significant (*β* = 0.360, *p* < 0.001), supporting H2; the effect of health awareness on actual behavioral control was statistically significant (*β* = 0.396, *p* < 0.001), supporting H3; the effect of health awareness on implementation intention was statistically significant (*β* = −0.327, *p* < 0.001), and supporting H4.

This study adopted the bootstrapping approach to verify the mediating effects [52]. Table 4 presents the results of 5000 bootstrap samples with a 95% confidence interval, with all Z values greater than 1.96 and no zeros within the 95% confidence intervals. In addition, significant mediation occurred between actual behavioral control and physical activity through implementation intention (the standardized indirect effect = 0.178, *p* < 0.01), which provides support to H5; significant mediation occurred between perceived policy effectiveness and physical activity through actual behavioral control and implementation intention (the standardized indirect effect = 0.337, *p* < 0.001), which provides support to H6; significant mediation occurred between health awareness and physical activity through actual behavioral control and implementation intention (the standardized indirect effect = 0.302, *p* < 0.001), which provides support to H7. The findings showed that working women with higher perceptions of policy effectiveness, more vital health awareness, higher actual behavioral control, and stronger implementation intentions are more likely to participate in physical activity.

## 6. Discussion

### 6.1. Contributions

This study makes the following contributions to the theoretical research of physical activity. First, the researchers constructed an extended TPB model to discuss the impact of policy factors and health awareness on working women’s participation in physical activity. The results showed that perceived policy effectiveness and health awareness significantly positively affect actual behavioral control and implementation intention. Perceived policy effectiveness and health awareness have the most decisive influence on actual behavioral control, followed by the impact on implementation intention. As shown in Figure 2, the variables can explain 50% of the variance in physical activity, which was much higher than the 20–30% variance in previous studies [53], indicating that the explanatory power of the extended TPB model has been dramatically improved. As the most classic theory in organizational behavior, the TPB is widely used in various fields. The researchers suggest that when applying TPB to study people’s behavior mechanisms, scholars should not only discuss the influence of psychological factors on behavior but also combine external environment and situational factors.

Second, this study is the first to verify the impact of perceived policy effectiveness on people’s participation in physical activity. The results suggested that actual behavioral control and implementation intention mediated the relationship between perceived policy effectiveness and physical activity. This shows that China’s “National Fitness Program” facilitates and creates favorable conditions for professional women to participate in sports activities. As an external factor that explains and predicts citizen behavior, policy support cannot be ignored. Researchers should consider the role of policies when studying organizational citizenship behavior and make recommendations for policymakers from the perspective of citizens’ perceptions and evaluations of the policies.

Third, this study is the first to demonstrate the relationship between health awareness and physical activity. The study’s results showed that the relationship between health awareness and physical activity was mediated by actual behavioral control and implementation intention, indicating that working women with more vital health awareness had stronger implementation intention and actively participated in physical activity. Health awareness is an overlooked factor among scholars exploring factors influencing people’s participation in physical activity. In real life, people who are more health conscious are more concerned about their health issues and are more willing to participate in activities that benefit their health.

### 6.2. Practical Implications

Considering the positive impact of perceived policy effectiveness on actual behavioral intentions and implementation intentions, the government should continue increasing policy support. With the continuous acceleration of urbanization, urban land is becoming increasingly tight, and there is a severe mismatch between urban population density and the coverage of public service facilities [54]. Even if working women are willing to participate in physical activity, they cannot find suitable venues for exercise, or the nearby sports venues (e.g., gyms) need to be paid to use; these constraints prevent working women from participating in physical activities. When physical activity has become a paid-for experience, this problem is also encountered by many ordinary people, especially those with low incomes and living in low-end communities. To improve the effectiveness of the policy, the government should not only increase financial investment, such as building more sports facilities and stadiums that people can use for free, but also strengthen the publicity of the policy, such by popularizing the benefits of fitness to the public and improving people’s fitness knowledge.

Health awareness is also one of the critical factors affecting working women’s participation in physical activity. The high-intensity work and fast-paced life in the city put many working women in a sub-healthy state, especially young women who are concerned about being promoted, making more money, and finding their favorite and suitable life partner. However, health is often the most overlooked. In this regard, governments and communities can help people improve their skills in making healthy choices by providing health information and health education. In addition, health literacy promotion can be enhanced through public service announcements, radio broadcasts, and banners. Furthermore, it can improve the collaboration between the government, communities, families, and individual residents and take effective intervention measures on health problems and influencing factors, forming a good situation in which the government is actively leading, and the society and individuals are widely involved.

## 7. Conclusions

In response to the proposed research objectives, this study pointed out that, under the influence of the “National Fitness Program”, about 35% of working women regularly or consistently participated in physical activities, indicating that the “National Fitness Program” is beginning to pay off. In addition, the results showed that perceived policy effectiveness, health awareness, and actual behavioral control are essential factors affecting working women’s implementation intention and participation in physical activity. In particular, perceived policy effectiveness and health awareness can directly affect implementation intention and indirectly affect participation in physical activity through the mediation of actual behavioral control and implementation intention. Therefore, this study recommends that the government deepen the breadth and depth of policy support and advocacy.

This study has certain limitations. First, nearly 50% of the respondents in this survey had a college/university degree or above, which affects the sample’s representativeness to a certain extent. Future research should focus more on health awareness and physical activity participation in low-income, less-educated women. Second, the sample loss rate in this study’s second stage of the questionnaire survey was too high. In future research, researchers should consider appropriately adding some material incentives (e.g., coffee coupons, shopping coupons) to increase the enthusiasm of respondents to participate in the survey. Third, this study did not ask respondents to report their physical activity levels when they first participated in the questionnaire. Therefore, it is recommended that follow-up research adopt the method of longitudinal analysis to compare respondents’ behavioral differences over time to explain the impact of independent variables on dependent variables more specifically. Fourth, this study’s relatively static research method has limitations in predicting individual behavior. Follow-up research can consider using the simulation method to reveal individuals’ behavior mechanisms dynamically.

## Figures and Tables

**Figure 1 ijerph-20-01021-f001:**
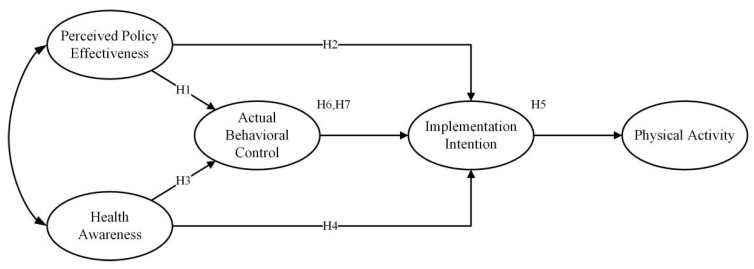
Conceptual model.

**Figure 2 ijerph-20-01021-f002:**
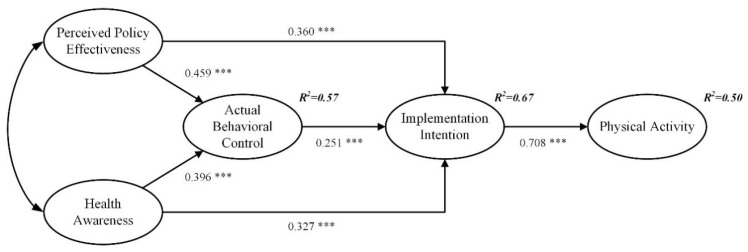
Structural path model. *** *p* < 0.001; standardized coefficients are reported.

**Table 1 ijerph-20-01021-t001:** Participant profile (N = 376).

Profiles	Survey	Census ^a,b^
** *Respondent age (%)* **		
≤28	36.7	15–59 (68.0%)
29–44	50.5	-
45–55	12.8	-
** *Respondent education level (%)* **		
Below high school	15.7	48.9
High school/Vocational school	35.6	21.3
College/University	32.7	29.8
Master or Ph.D.	16.0	
** *Respondent occupation (%)* **		
Public sector	30.3	33.4
Private sector	69.7	66.6
** *Respondent monthly salary (%)* **		
≤5000 CNY	35.2	Mean 5677 CNY
5001–10,000 CNY	29.4	
10,001–20,000 CNY	23.0	
≥20,001 CNY	12.3	

^a,b^ Changsha Municipal Statistics Bureau [45,46].

**Table 2 ijerph-20-01021-t002:** Reliability and validity tests.

Items	Loadings	Cα	AVE	CR
** *Perceived policy effectiveness* **		0.869	0.571	0.869
PPE1: The Government has increased financial investment to support the “National Fitness Program”.	0.755			
PPE2: The Government campaign helps citizens understand the importance of physical activity.	0.746			
PPE3: The Government campaign clearly explains the benefits of physical activity.	0.747			
PPE4: The Government promotes physical activity as a positive symbol, label, image, and event.	0.787			
PPE5: The Government’s policy facilitates me to participate in more physical activities.	0.742			
** *Health awareness* **		0.897	0.636	0.897
HA1: I reflect on my health a lot.	0.779			
HA2: I am very self-conscious about my health.	0.810			
HA3: I am constantly examining my health.	0.832			
HA4: I am alert to changes in my health.	0.796			
HA5: I am aware of the state of my health as I go through the day.	0.770			
** *Implementation intention* **		0.907	0.765	0.907
IMP1: During the next week, I clarified what types of physical activity I would participate in.	0.864			
IMP2: During the next week, I clarified which days and times I would participate in physical activity.	0.881			
IMP3: During the next week, I clarified where I would participate in physical activity.	0.880			
** *Actual behavioral control* **		0.848	0.659	0.853
ABC1: During the previous week, I was able to exercise as I planned.	0.820			
ABC2: During the previous week, I was able to find a place to exercise.	0.824			
ABC3: During the previous week, I was confident about my fitness knowledge/skills.	0.792			
** *Physical activity* **		0.924	0.805	0.925
PA1: During the previous week, how often did you engage in light physical activity such as walking, light cleaning, ranking lawn, or lightly strenuous exercise such as yoga, bowling or similar activities?	0.902			
PA2: During the previous week, how often did you engage in gardening, carrying loads upstairs or moderately strenuous sports such as gymnastics, swimming, bicycling, strength conditioning or similar activities?	0.912			
PA3: During the previous week, how often did you engage in strenuous sport and conditioning exercise such as running, jogging, soccer, tennis, aerobics or similar activities?	0.877			

All standardized loadings are significant at the 0.001 level.

**Table 3 ijerph-20-01021-t003:** Discriminant validity test.

Construct	PPE	HA	IMP	ABC	PA
PPE	**(0.756)**				
HA	0.509 **	**(0.797)**			
IMP	0.631 **	0.632 **	**(0.875)**		
ABC	0.588 **	0.576 **	0.623 **	**(0.812)**	
PA	0.514 **	0.462 **	0.639 **	0.476 **	**(0.897)**

The square root of the average variance extracted (AVE) is in diagonals (bold); off diagonals are Pearson’s correlations of constructs. ** *p* < 0.01.

**Table 4 ijerph-20-01021-t004:** Standardized indirect effects.

	Point Estimate	Product of Coefficients		Bootstrapping				
				Percentile 95% CI		Bias-Corrected 95% CI		Two-Tailed Significance
		*SE*	*Z*	Lower	Upper	Lower	Upper	
Indirect effects								
ABC → PA	0.178	0.061	2.918	0.057	0.302	0.056	0.302	0.004 (**)
PPE → PA	0.337	0.047	7.170	0.243	0.429	0.243	0.429	0.000 (***)
HA → PA	0.302	0.044	6.864	0.212	0.386	0.218	0.391	0.000 (***)

Standardized estimation of 5000 bootstrap samples; ** *p* < 0.01, *** *p* < 0.001.

## Data Availability

Not applicable.

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
