# Peer review of "Promoting Physical Activity among Working Women: The Influence of Perceived Policy Effectiveness and Health Awareness"

_ijerph, 2023, doi:10.3390/ijerph20021021_

Round 1
Reviewer 1 Report (Previous Reviewer 2)
I cant understend this results: In the first stage, 800 questionnaires were distributed, 662 valid questionnaires were retrieved, 235 and the recovery rate of valid questionnaires was 82.8%. In the second stage, 662 questionnaires were distributed, and 376 valid questionnaires were recovered. As a result, the recovery rate of valid questionnaires was 56.8%". What was wrong? What was wrong? Why didn't the authors try to collect more missing questionnaires? Unfortunately, this casts doubt on the correctness of the conclusions. This is the weakness of that manusript.
Author Response
Please see the attachment. Thank you.

Reviewer 2 Report (New Reviewer)
The manuscript by Wang and colleagues, titled "Promoting physical activity among working women: the influence of perceived policy effectiveness and health awareness," explores the impact of perceived policy effectiveness and health awareness, elicited by the "National Fitness Program" in China, on the actual behavioral control and implementation intention of physical activity by working women in five municipal districts of Changsha City. Adopting a longitudinal (1-week) research method, a two-stage survey was conducted. The study design is very clear and all hypotheses are well supported. The whole study is well executed and rigorous statistical analysis was carried out.
My main concerns about this study are only two:
1. In the scenario described by the authors what is missing is the starting point, which could render all efforts useless: How can they claim that the obtained results are related to the National Fitness Program claims if they do not report the participants' PA level at the study baseline as well as their weekly sport frequency, if any?
2. And so, how can the results obtained after one week represent a stable behavioral change in the daily patterns of working women?
Author Response
Please see the attachment. Thank you.

This manuscript is a resubmission of an earlier submission. The following is a list of the peer review reports and author responses from that submission.
Round 1
Reviewer 1 Report
The idea for this research is good but not in its present form.
It is not clear what the main point of this paper is. All content must be directed and connected to the aim of the research.
There seems to be a lot of irrelevant content throughout the paper. It is not easy to follow the paper due to its structure and language.
The title needs to be rephrased according to the aim. The abstract should also be improved in terms of main information and style.
The structure and style of the paper is poor. The purpose, literature review and design need to support one another in a straightforward and logical manner.
The statistical analyses are correct. But it is unclear what role the profile of participants play in understanding the results and the value the main points/discussion.
The 6.1 is over-emphasizing the contribution of the paper. The language needs to be refined.
Reviewer 2 Report
The entire manuscript is very well written. A proper introduction introduces you to the state of knowledge. My attention was focused on the methodology of this research. There are no conditions for inclusion and exclusion from research. How were the respondents selected? Randomization? How? Why were the losses of the questionnaires so different? It posible that answers come only from satisfied part of society. Can you get reliable data from such satisfaction surveys?
Round 2
Reviewer 2 Report
Thanks for any clarifications. I can see that the text of the manuscript is very well prepared. On the other hand, I have doubts about the correctness of conclusions based on such studies, such subjects. It is only my doubt.